# Sucrose-Based Screening of a Novel Strain, *Limimaricola* sp. YI8, and Its Application to Polyhydroxybutyrate Production from Molasses

**DOI:** 10.3390/polym17111471

**Published:** 2025-05-26

**Authors:** Yeda Lee, Dohyun Cho, Yuni Shin, Yebin Han, Gaeun Lim, Jongmin Jeon, Jeongjun Yoon, Jeongchan Joo, Hwabong Jeong, Jungoh Ahn, Shashi Kant Bhatia, Yunghun Yang

**Affiliations:** 1Advanced Materials Program, Department of Biological Engineering, College of Engineering, Konkuk University, 120, Neungdong-ro, Gwangjin-gu, Seoul 05029, Republic of Korea; yeyeyda@gmail.com (Y.L.); ehguswh1997@naver.com (D.C.); sdbsdl0526@naver.com (Y.S.); hanyebin3@gmail.com (Y.H.); lge0919@naver.com (G.L.); shashikonkukuni@konkuk.ac.kr (S.K.B.); 2Green & Sustainable Materials Research and Development Department, Korea Institute of Industrial Technology (KITECH), Cheonan 31056, Republic of Korea; j2pco@kitech.re.kr (J.J.); jjyoon@kitech.re.kr (J.Y.); 3Department of Systems Biotechnology, Chung Ang University, Anseong 17546, Republic of Korea; 4Department of Chemical Engineering, Kyung Hee University, Giheung-gu, Yongin-si 17104, Republic of Korea; 5Biotechnology Process Engineering Center, Korea Research Institute Bioscience Biotechnology (KRIBB), Jeongeup-si 580-185, Republic of Korea; jhb9951@kribb.re.kr (H.J.); ahnjo@kribb.re.kr (J.A.); 6Applied Biological Engineering, University of Science and Technology, 217 Gajeong-ro, Yuseong-gu, Daejeon 32113, Republic of Korea; 7Institute for Ubiquitous Information Technology and Applications, Konkuk University, Seoul 05029, Republic of Korea

**Keywords:** polyhydroxybutyrate, *Limimaricola* sp. YI8, *Escherichia coli*, *Cupriavidus necator*, gel permeation chromatography, universal test machine, differential scanning calorimetry

## Abstract

Poly(3-hydroxybutyrate) is a biodegradable plastic produced by various microbes. Considering the emerging environmental problems caused by plastics, P(3HB) has gained attention as a substitute for conventional plastics. In this study, we isolated a novel P(3HB)-producing microbe, *Limimaricola* sp. YI8, which utilized sucrose as a cost-effective carbon source for P(3HB) production. Under optimized conditions, *Limimaricola* sp. YI8 produced 6.2 g/L P(3HB) using sucrose as the sole carbon source. P(3HB) extracted from YI8 exhibited a pinkish color derived from a dye produced naturally by YI8. Films fabricated from extracted P(3HB) polymer were subjected to analyses, including gel permeation chromatography, universal test machine, and differential scanning calorimetry, to determine their physical properties. The obtained values were almost identical to those of P(3HB) films extracted from *Escherichia coli* and *Cupriavidus necator* H16. Overall, this study presents the potential of *Limimaricola* spp. YI8 as a P(3HB)-producing strain and the P(3HB) films extracted from this strain.

## 1. Introduction

Poly(3-hydroxybutyrate) (P(3HB)) is a biodegradable plastic with properties similar to those of conventional plastics [1]. Owing to increasing environmental concerns regarding conventional petroleum-based plastics, P(3HB) has gained attention as a suitable substitute [2,3]. The P(3HB) can accumulate inside microbial cells in the presence of an excess carbon source and limited amounts of other nutrients, such as nitrogen and phosphorous [1]. P(3HB) accumulation varies among different microorganisms, and some species, such as *Cupriavidus necator*, *Halomonas* sp., and *Bacillus* sp., are well known as P(3HB) producers [2,4,5,6,7].

High cost is one of the major challenges in scaling up P(3HB) production for industrial applications [8]. Several methods have been explored to reduce production costs, among which the use of cheaper substrates is the most efficient and simplest approach [2,3,8]. Sugarcane is among the most abundant resources worldwide, and molasses derived from sugar extraction is relatively inexpensive, making it an attractive carbon source for the mass production of P(3HB) [9,10,11]. In molasses concentrated from sugarcane juice, sucrose accounts for approximately 35% of the total carbohydrates. It also contains various nutrients, including nitrogenous compounds, vitamins, and sugars. Therefore, numerous efforts have been made to utilize molasses as a low-cost carbon source in fermentation processes [12,13].

However, efficient utilization of sucrose as a carbon source for P(3HB) production depends on the inherent metabolic capabilities of microorganisms, which can vary among different species [14]. When a P(3HB)-producing strain is unable to metabolize sucrose effectively, metabolic engineering can be employed to confer this ability [10]. *Cupriavidus necator* H16, a promising candidate strain for P(3HB) production, also cannot metabolize sucrose well; therefore, several studies have reported relevant metabolic engineering approaches to enhance sucrose utilization for P(3HB) production [15,16,17,18,19,20,21,22] (Table 1).

In the present study, a novel strain, *Limimaricola* sp. YI8, which has an inherent metabolic capacity to utilize sucrose well for P(3HB) production, was isolated, and optimization experiments were conducted using sucrose as a carbon source [23,24]. Furthermore, in order to systematically evaluate the effects of individual medium components on cell growth and P(3HB) yield, a design of experiments (DOE) approach was employed. This statistical method enabled the identification of key nutritional factors and their interactions influencing microbial performance, thereby providing valuable insight into the medium optimization process for this newly identified strain. To the best of our knowledge, there have been no previous reports of P(3HB) production by this species, making this study the first to investigate the potential of *Limimaricola* sp. YI8 for P(3HB) production.

## 2. Materials and Methods

### 2.1. Chemical Reagents

All media components used in this study were acquired from BD Difco Laboratories (Becton-Dickinson, Franklin Lakes, NJ, USA). Reagents for gas chromatography (GC) and high-performance liquid chromatography (HPLC) (chloroform, methanol, and other derivatization reagents) were purchased from Sigma-Aldrich (St. Louis, MO, USA). Glucose was purchased from Duksan Pure Chemical (Seoul, Republic of Korea). Sugarcane molasses was purchased from Evermiracle (Jeonju, South Korea).

### 2.2. Bacterial Strain and Culture Conditions for P(3HB) Synthesis

The strains and plasmids used in the experiments are listed in Table 2. To assess the characteristics of P(3HB) produced by *Limimaricola* sp. YI8 in comparison with representative industrial strains, *Cupriavidus necator* H16 and *Escherichia coli*::pLW487 were cultured under defined conditions. *Cupriavidus necator* H16 was cultivated in 5 mL of ReMM medium, inoculated at 1% (*v*/*v*), and incubated at 30 °C with shaking at 200 rpm for 72 h. The ReMM medium contained 20 g/L NaH_2_PO_4_, 23 g/L Na_2_HPO_4_, 2.25 g/L K_2_SO_4_, 0.39 g/L MgSO_4_, 0.062 g/L CaCl_2_, and trace elements (15 mg/L FeSO_4_·7H_2_O, 2.4 mg/L MnSO_4_·H_2_O, 2.4 mg/L ZnSO_4_·7H_2_O, and 0.48 mg/L CuSO_4_·5H_2_O) dissolved in 0.1 M HCl. Fructose was added at 2% (*w*/*v*) as the sole carbon source.

*Escherichia coli* harboring pLW487 plasmids strains were cultivated in 10 mL of M9 medium at 30 °C for 96 h. The M9 medium consisted of 6.0 g/L Na_2_HPO_4_, 3.0 g/L KH_2_PO_4_, 0.5 g/L NaCl, 1.0 g/L NH_4_Cl, 1 mM MgSO_4_, and 0.1 mM CaCl_2_.

### 2.3. Bacterial Isolation and Plate Assay for Identifying P(3HB)-Producing Strain

The strain *Limimaricola* sp. YI8 was isolated from the coast of Anmyondo in South Korea. Seawater samples were appropriately diluted with distilled water to a dilution factor of 100 to 1000 and spread on Marine Broth (MB; Difco) agar plates supplemented with 1% (*w*/*v*) sucrose. The plates were then incubated at 30 °C for 2 days, and colonies displaying various morphologies were isolated. Each colony was subsequently cultured in 5 mL of MB medium for 1 day; the culture broth was then mixed with sterile 20% (*w*/*v*) glycerol and stored at −81 °C until further use.

Subsequently, Sudan black B staining was used to identify P(3HB)-producing strains [27]. The isolated bacteria were spread on MB agar plates supplemented with 1% sucrose and incubated at 30 °C for 2 days. The plates were stained with a 0.02% (*w*/*v*) Sudan Black B ethanolic solution for 20 min and washed with 96% ethanol to remove the dye. Colonies stained with Sudan Black B were considered positive for P(3HB) production.

To identify the species of P(3HB)-producing strains, 16S rRNA sequencing was performed using polymerase chain reaction using the primer 27F. Partial sequences were obtained from Bionics (Seoul, South Korea). These sequences were compared with the entries in the GenBank database of the National Center for Biotechnology Information using BLAST (https://blast.ncbi.nlm.nih.gov/Blast.cgi, accessed on 9 July 2024).

### 2.4. Evaluation of Carbon Sources Utilization for P(3HB) Production

To identify the preferred carbon source of *Limimaricola* sp. YI8, eight carbon sources (glucose, fructose, xylose, glycerol, sucrose, lactose, lactate, galactose) were added at 1% (*w*/*v*) for P(3HB) production. The pre-culture was conducted in 5 mL of MB medium (peptone 5 g/L, yeast extract 1 g/L, ferric citrate 0.1 g/L, sodium chloride 19.45 g/L, magnesium chloride 5.9 g/L, sodium sulfate 3.24 g/L, calcium chloride 1.8 g/L, potassium chloride 0.55 g/L, sodium bicarbonate 0.16 g/L, potassium bromide 0.08 g/L, strontium chloride 0.034 g/L, boric acid 0.022 g/L, sodium silicate 0.004 g/L, sodium fluoride 0.0024 g/L, ammonium nitrate 0.0016 g/L, and disodium phosphate 0.008 g/L) in a 14 mL test tube and incubated at 30 °C with shaking at 200 rpm. After 24 h, 5 mL of MB medium was inoculated with the 1% (*w*/*v*) pre-culture broth, and 1% (*w*/*v*) of each type of sugar was added as a carbon source. The cultures were incubated at 30 °C with shaking at 200 rpm for 72 h. After incubation, the cells were harvested by centrifugation at 4500 rpm and 25 °C for 20 min. The collected cells were used for the analysis of dry cell weight and P(3HB) production. All experiments were conducted independently and in duplicate.

### 2.5. Transmission Electron Microscopy Analysis

To observe the intracellular accumulation of P(3HB) granules in *Limimaricola* sp. YI8 by transmission electron microscopy (TEM), cells were inoculated at 1% (*v*/*v*) into 5 mL of Marine Broth (MB) medium and incubated at 30 °C with shaking at 200 rpm for 72 h. After cultivation, 1 mL of the culture was harvested by centrifugation at 13,000 rpm for 3 min, and the resulting cell pellet was collected for TEM analysis.

The collected cell pellet was fixed with Karnovsky’s solution containing 2% glutaraldehyde. Post-fixation was performed using 1% osmium tetroxide in 0.05 M sodium cacodylate buffer. The sample was dehydrated stepwise using a graded ethanol series of 50%, 70%, 95%, and 100% (*v*/*v*), with each step lasting approximately 10 min. Following dehydration, the sample was treated with propylene oxide to serve as a transitional solvent. Infiltration was performed by sequentially incubating the sample in mixtures of propylene oxide and Spurr’s resin at volume ratios of 1:1 and 1:2. Finally, the sample was embedded in 100% Spurr’s resin. Polymerization was conducted at 70 °C in a dry oven. Ultrathin sections were prepared using an ultramicrotome (LEICA, EM UC7), mounted on TEM grids, and observed with an EF-TEM (Carl Zeiss, LIBRA 120) operated at 120 kV.

### 2.6. Statistical Analysis

Minitab 21 (Minitab Inc., State College, PA, USA) was used for the experimental design and subsequent regression analysis of the experimental data. Significant differences were determined using analysis of variance. The quality of the polynomial model equation was judged statistically using the coefficient of determination R^2^, and its statistical significance was determined using an F-test. The significance of regression coefficients was tested using t-tests. In this case, yeast extract, sodium chloride, and sucrose significantly affected P(3HB) yield (*p* < 0.05).

### 2.7. Screening of Essential Medium Components Using the Plackett–Burman Design

The Plackett–Burman design was employed to identify significant factors. A total of 20 experiments were performed in duplicate to assess 10 variables. Ten independent variables (k = 10) were selected for this study, each tested at two levels: high (+) and low (−). Additionally, two dummy variables were included in the 20 trials (3.c) to estimate the standard error.

### 2.8. Box-Behnken Design and Response Surface Methodology

The significant variables identified in the previous experiments were optimized using the Box–Behnken design at different levels (4.c), whereas non-significant variables were maintained at their initial medium levels. A regression equation was developed to describe the relationship between the coded and actual values, as shown in Equation (1) below.(1)$$X_{i}=\frac{\left(A_{i}-A_{0}\right)}{∆A_{i}}$$
where $X_{i}$ represents the coded value of the *i*th variable, $A_{i}$ is its actual value, $A_{0}$ is the actual value at the center point, and ∆A denotes the step change of the *i*th variable. The correlation between the response and the four variables was modeled using a predictive quadratic polynomial equation:(2)$$Y=\beta_{0}+\Sigma \beta_{i}X_{i}+{\Sigma \beta }_{ij}X_{i}X_{j}+{\Sigma \beta }_{ii}X_{i}^{2}i=1,2,3,\cdots \cdots k$$

Here, Y represents the predicted response, $\beta_{0}$ is the intercept term, $\beta_{i}$ denotes the linear coefficient, $\beta_{ii}$ is the squared coefficient, and $\beta_{ij}$ is the interaction coefficient. The independent factors (medium components) are expressed as coded values $X_{i}$ and $X_{j}$. The accuracy and predictive capability of this polynomial model were assessed using the coefficient of determination ($R^{2}$).

Each experiment was conducted in duplicate, and the mean values were used for further analyses. Variables that were statistically significant at a 95% confidence level (*p* < 0.05) in the regression analysis were considered to have a substantial effect on cell growth and P(3HB) production. These significant variables were further optimized using response surface methodology with the Box-Behnken design.

### 2.9. P(3HB) Production Using Molasses

Sucrose in molasses may break down into glucose or fructose at high temperatures; therefore, filtration rather than autoclaving was used for sterilization. Molasses was diluted to 20% with distilled water and centrifuged at 13,000 rpm at 25 °C for 20 min. The obtained supernatant was adjusted to pH 7 using 10 N NaOH and 5 N HCl and then filtered with a 28 mm syringe filter through a 0.22 μm polyethersulfone membrane (Sartorious, Göttingen, Germany). Diluted molasses was added at different concentrations (1, 2, 2.5, 3, 3.5, and 4%) to 5 mL of MB medium. Pre-cultured medium [1% (*v*/*v*)] was inoculated, and the main culture medium was incubated at 30 °C, 200 rpm for 3 days.

### 2.10. Analytical Methods

#### 2.10.1. HPLC

To measure the carbohydrate contents in molasses, diluted molasses (0.5%) was assayed by HPLC with an Aminex HPX-87H column (BioRad, Hercules, CA, USA), coupled to an ultraviolet (UV at 210 nm) and refractive index (RI) detector, using 5 mM sulfuric acid (H_2_SO_4_) as the eluent, at a flow rate of 0.600 mL min^−1^ and an oven temperature of 50 °C.

#### 2.10.2. GC

P(3HB) was quantified using GC (Young-lin Tech, Seoul, Republic of Korea) [28]. For sample preparation, the culture broth was centrifuged to separate cell pellets from the supernatant. Cell pellets were washed twice with deionized water, transferred to glass vials, and lyophilized. Lyophilized cells were used to measure dry cell weight (DCW). Methanolysis was conducted for GC analysis. Briefly, 1 mL of chloroform and 1 mL of 15% (*v*/*v*) H_2_SO_4_/85% methanol solution were added to the lyophilized cells and heated at 100 °C for 2 h. After reaction completion, the samples were cooled to room temperature. Subsequently, 1 mL of deionized water was added to each sample and vortexed twice for 10 s. The chloroform layer was carefully transferred to an Eppendorf tube, and the residual water was removed using anhydrous crystalline Na_2_SO_4_. Sample preparation was completed by filtering the samples into GC vials using a 0.22 μm Millex-GP syringe filter. The samples were injected into a model 6500 gas chromatograph with split mode (1/10) (Young-lin Tech, Seoul, Republic of Korea) equipped with a fused silica capillary column (Agilent HP-FFAP, 30 m × 0.32 mm, i.d. 0.25-μm film) and a flame ionization detector (FID). The inlet temperature was set at 210 °C, and helium was used as the carrier gas at a flow rate of 3 mL min^−1^. The oven temperature followed a gradient program: 80 °C from 0 to 5 min and 220 °C from 12 to 18 min. During the operation, the temperature of the FID was maintained at 230 °C.

### 2.11. P(3HB) Extraction and Characterization

P(3HB) was extracted from the cells using solvent extraction [29]. *Limimaricola* sp. YI8 was cultured in 50 mL of MB broth containing 3% (*w*/*v*) sucrose in a 250 mL baffled flask at 30 °C, 200 rpm for 72 h. Then the culture broth was centrifuged at 4500 rpm at 25 °C for 20 min, washed twice with deionized water, followed by lyophilization. Further, 50 mL of chloroform was added to dried cell pellets and placed into a 60 °C water bath for 6 h. To remove cell debris, the P(3HB)-containing chloroform was filtered through a Whatman No. 1 filter paper, dropped onto a Petri dish, and left at room temperature to allow chloroform evaporation.

### 2.12. Identification of the Mechanical and Thermal Properties

The number-average molecular weight (M_n_), weight-average molecular weight (M_w_), and polydispersity index (PDI) of each film were determined by gel permeation chromatography (GPC; YOUNGIN Chromass, Seoul, Republic of Korea). The chromatograph was equipped with a loop injector (Rheodyne 7725i), isocratic pump with dual heads (YL9112), column oven (YL9131), columns (K-G 4 A, guard column; K-804 8.0 × I.D. × 300 mm; K-805, 8.0 × 300 mm; Shodex), and refractive index detector (YL9170) [30,31]. Sample preparation was conducted by melting 10 mg of P(3HB) films in chloroform at 60 °C for 2 h, followed by filtering using a 0.2 μm syringe filter to remove impurities. As a mobile phase, chloroform was used at a 1.0 mL min^−1^ flow rate, and the oven temperature was maintained at 40 °C. The injection volume of the samples was 60 μL, and the split ratio was 3:1.

The mechanical properties of the P(3HB) films were determined using a Universal Testing Machine (UTM) (EZ-SX; Shimadzu, Kyoto, Japan) [30]. Sample preparation was conducted by cutting the P(3HB) films into rectangular shapes with dimensions of 60 × 10 mm (length × width). The thicknesses of P(3HB) film samples were measured using a digital caliper 1111–100 A (Insize, Loganville, GA, USA). The strain rate was set at 20 mm min^−1^. TRAPEZIUM X software was used to calculate the tensile strength, elongation at break, and Young’s modulus.

To analyze the thermal properties of PHA, differential scanning calorimetry (DSC 4000; Perkin Elmer, Shelton, CT, USA) was performed at temperatures ranging from 60 °C to 200 °C [30]. The rate of temperature increase was set at 10 °C min^−1^ during the heating process and was reversed during the cooling process. All processes were conducted under a N_2_ atmosphere at a flow rate of 20-cc min^−1^.

## 3. Results and Discussion

### 3.1. Screening and Isolation of the P(3HB)-Producing Strain Limimaricola sp. YI8

To identify the P(3HB)-producing strain, diluted seawater samples were plated on MB agar plates containing 1% sucrose, and the resulting colonies were stained with Sudan Black B. Stained colonies were considered positive for P(3HB) production and cultivated in 5 mL of MB media containing 1% sucrose at 30 °C for 3 days. Cells were harvested by centrifugation at 4500 rpm at 25 °C for 20 min, and the cell pellets were washed twice with deionized water, followed by lyophilization. DCW and P(3HB) production in each strain were determined (Figure 1A).

Among nine colonies, YI8 showed the highest P(3HB) production (1.8 g/L) with relatively high cell growth (4 g/L). After identifying the P(3HB)-producing ability of the YI8 strain, it was subjected to 16s rRNA sequencing, and a phylogenetic tree was constructed (Figure 1B). Based on 16s rRNA sequencing and phylogenetic tree analysis, YI8 was identified as belonging to the *Limimaricola* species, showing a 99.67% relatedness with *Limimaricola variabilis* J-MR2-Y, previously reported as an oil degrading consortium [32]. Consequently, we designated YI8 as *Limimaricola* sp. YI8 and selected this strain for further P(3HB) production experiments.

We also tested the ability of *Limimaricola* sp. YI8 strain to utilize eight different carbon sources—glucose, fructose, xylose, glycerol, sucrose, lactose, lactate, and galactose—each supplemented at 1% (*w*/*v*) in 5 mL of MB medium (Figure 2A). The results revealed that YI8 efficiently utilized glycerol and sucrose as carbon sources. Using glycerol as the carbon source resulted in a cell mass of 4.8 g/L and P(3HB) production of 2.2 g/L. In contrast, when sucrose was used as the carbon source, the cell mass was 4.5 g/L, and P(3HB) production reached 2.3 g/L. Although glycerol led to a slightly higher cell mass, sucrose resulted in a marginally better P(3HB) production ratio of 52% compared to that with glycerol (48%). Therefore, sucrose, which is a major component of molasses, was determined as the most suitable carbon source for P(3HB) production. The accumulation of P(3HB) granules within the cells was confirmed by transmission electron microscopy (Figure 2B).

### 3.2. Screening of Key Medium Components Using the Plackett–Burman Design

As *Limimaricola* sp. YI8 was newly isolated from the marine environment, and no efficient media were reported for P(3HB) production, the variables were selected based on Marine Broth medium, which provides the necessary salts and nutrients for the growth of marine microorganisms adapted to high salinity [7]. MB contains 5 g/L peptone, 1.0 g/L yeast extract, 0.1 g/L ferric citrate, 19.45 g/L sodium chloride, 5.9 g/L magnesium chloride, 3.24 g/L magnesium sulfate, 1.8 g/L calcium chloride, 0.55 g/L potassium chloride, 0.16 g/L sodium bicarbonate, 0.08 g/L potassium bromide, 0.034 g/L strontium chloride, 0.022 g/L boric acid, 0.004 g/L sodium silicate, 0.0024 g/L sodium fluoride, 0.0016 g/L ammonium nitrate, and 0.008 g/L disodium phosphate. Among these, nine substances with the highest addition levels were set as variables.

The Plackett–Burman design was employed for the initial screening of the medium components. Ten different components, including carbon and nitrogen sources (peptone, yeast extract, and sucrose) and inorganic salts (sodium chloride, MgCl_2_, MgSO_4_, CaCl_2_, K_2_SO_4_, NaHCO_3_, and boric acid), were assessed for their ability to enhance P(3HB) production by *Limimaricola* sp. YI8. The effects of these ten components are presented in Appendix A. Consequently, three components—peptone, yeast extract, and sucrose—significantly impacted cell growth and P(3HB) production (Figure 3A,B). In contrast, MgCl_2_, MgSO_4_, K_2_SO_4_, NaHCO_3_, and boric acid showed no significant effect on cell growth or P(3HB) production. Subsequently, the exact optimal values for these factors were determined using a Box–Behnken design.

### 3.3. Optimization of Medium Components Using the Box–Behnken Design

The three significant factors were examined using a Box–Behnken design with 15 runs (Appendix A). The cell mass and P(3HB) production were assessed to determine the combined effects of these factors within their specific ranges. Variables that showed significant effects with confidence levels (>95%) in the Plackett–Burman design were chosen for further optimization using the Box–Behnken design. A contour plot was generated, revealing that yeast extract had a positive effect on cell growth but tended to decrease the P(3HB) content. Sucrose showed positive results only on P(3HB) production but had little effect on the DCW. In contrast, sodium chloride was found to negatively affect both cell growth and P(3HB) production, highlighting its detrimental impact on the overall process (Figure 4). The anticipated maximum P(3HB) production and cell growth were 0.55 g/L and 3.045 g/L, respectively, using media containing 5 g/L of sucrose and 1.71 g/L of yeast extract, without NaCl.

### 3.4. Examination of P(3HB) Production Using Molasses as a Carbon Source

Molasses is rich in sugars such as sucrose, glucose, and fructose, with sucrose being the most abundant, accounting for approximately 35% [12,13]. As molasses is a complex and variable by-product of sugar production, its composition can vary depending on the source and processing conditions. In this study, commercially available sugarcane molasses (Evermiracle, Jeonju, Korea) was used. According to the sugar composition analysis, the molasses used in this study contained approximately 58.0% sucrose, 23.3% fructose, and 18.7% glucose. Given the efficient utilization of sucrose by *Limimaricola* sp. YI8, we evaluated the potential of using molasses as a substitute for sucrose during P(3HB) production (Figure 5).

When *Limimaricola* sp. YI8 was cultured in 5 mL MB medium with molasses as the carbon source at different concentrations (1, 2, 2.5, 3, 3.5, and 4%), the maximum biomass and P(3HB) production were achieved at 3% (sucrose 11.65 g/L), yielding 8.5 g/L of DCW and 5.2 g/L of P(3HB) production. When using molasses instead of sucrose as a carbon source, both cell mass and P(3HB) content increased significantly compared to when sucrose was used. Considering the manufacturing cost and complexity associated with genetic engineering to improve P(3HB)-producing strains, this result indicates a positive impact for P(3HB) production using *Limimaricola* sp. YI8.

### 3.5. P(3HB) Film Preparation from Limimaricola sp. YI8 Cultivation

The formation of P(3HB) films was evaluated by solvent extraction using chloroform. *Limimaricola* sp. YI8 was cultured in 50 mL MB medium containing 3% sucrose. DCW and P(3HB) production improved in optimized cultivation at 11.1 and 6.2 g/L. After the extraction of P(3HB) and formation of the P(3HB) film, the pinkish color of the cells was co-extracted with P(3HB) and presented in the film, resulting in a pinkish-colored P(3HB) film being formed.

We then compared the color of the P(3HB) film obtained from YI8 with P(3HB) films obtained from other strains: wild-type *Cupriavidus necator* H16 and *Escherichia coli* KSYH harboring a P(3HB)-producing operon. P(3HB) production using *E. coli* was carried out with *E. coli* KSYH harboring pLW487, a pEP2-based plasmid containing the P(3HB) synthesis genes β-ketothiolase (*bktB*), acetoacetyl-CoA reductase (*phaB*), and polyhydroxyalkanoate synthase (*phaC*) from *Cupriavidus necator* [7,25,26]. Cultivation of *Escherichia coli* and *Cupriavidus necator* for P(3HB) production was performed using optimal conditions established in previous reports [7,33,34]. To briefly describe the conditions used in this study, *C. necator* was cultured in 5 mL of ReMM medium containing phosphate, sulfate, magnesium, calcium, and trace elements, with 2% (*w*/*v*) fructose at 30 °C and 200 rpm for 72 h. *E. coli* KSYH was grown in 10 mL of M9 medium with 2% (*w*/*v*) starch, 0.15% (*w*/*v*) yeast extract, 10 μM IPTG, and 10 mM glycine betaine at 30 °C for 96 h.

A visual comparison of the P(3HB) films obtained from each strain is depicted in Figure 6. The P(3HB) film from *Limimaricola* sp. YI8 showed a distinct pinkish color, which could aid visualization. In a previous report, *Limimaricola hongkongensis* and *Limimaricola variabilis* also formed pinkish colonies, considered a result of carotenoid pigments [14]. The pigments protect cells from direct sunlight, thereby safeguarding the bacterial culture from damage caused by ultraviolet light exposure [35]. In contrast, P(3HB) films obtained from *Escherichia coli* and *Cupriavidus necator* were colorless and yellowish, respectively.

### 3.6. Mechanical and Physical Properties of P(3HB) Films Obtained from Limimaricola sp. YI8

GPC analysis was performed to identify the number of average molecular weight (M_n_), average molecular weight (M_w_), and PDI (M_w_/M_n_) of each P(3HB) film (Table 3). The PDI of H16-P(3HB) (1.28 ± 0.01) was slightly higher than that of YI8-P(3HB) (1.24 ± 0.03), indicating that YI8-P(3HB) has a slightly more uniform molecular weight distribution [27]. The values obtained for *Limimaricola* sp. YI8 were similar to those of *E. coli* and *C. necator* H16 as well as of other wild-type P(3HB)-producing bacteria reported in previous studies [36].

UTM analysis was performed using P(3HB) films extracted from *Limimaricola* sp. YI8, *E. coli*, and *C. necator* H16. The values of tensile strength, elongation at break, and Young’s modulus obtained from each P(3HB) film were compared (Table 3). Comparison of the mechanical properties of the P(3HB) films showed that the tensile strength of the *Limimaricola* sp. YI8 P(3HB) film (32.5 MPa) was slightly lower than that of the *E. coli* (36.4 MPa) and higher than that of the *C. necator* H16 (22.1 MPa) films. However, elongation at break was the lowest (3%) compared to that of the *E. coli* (7.2%) and *C. necator* H16 (12.9%) films. Further, the Young’s modulus of the YI8-P(3HB) film was higher (1931.9 MPa) than those of the *E. coli* (1337.6 MPa) and *C. necator* H16 (701.7 MPa) films. Based on these results, P(3HB) obtained from *Limimaricola* sp. YI8 was determined to be relatively more rigid and less flexible than that from *E. coli* and *C. necator* H16 [35,37].

Even with the same P(3HB), thermal properties can vary depending on the microbial strain. Therefore, DSC analysis was conducted on the P(3HB) films obtained from *Limimaricola* sp. YI8, *E. coli*, and *C. necator* H16. Crystallization temperature (Tc) and melting point (Tm) values obtained from DSC analysis are listed in Table 4 and Appendix A. The melting temperature of all samples was nearly identical, approximately 172 °C. These values are similar to those previously reported for standard P(3HB) films [38]. YI8-P(3HB) showed a crystallization temperature (Tc) of 106.2 °C, which was significantly higher than that of H16-P(3HB) (84.7 °C). Typically, a lower Tc value indicates slower crystallization and a higher proportion of amorphous regions during cooling, leading to increased flexibility. In contrast, the higher Tc of YI8-P(3HB) suggests faster crystallization and a more crystalline structure, resulting in increased hardness. Consequently, the elevated modulus and Tc of YI8-P(3HB) contribute to its enhanced rigidity and physical stability, particularly at elevated temperatures, making it more robust and less prone to deformation.

Incorporating a dye may reduce the transparency of P(3HB), thereby affecting its optical properties such as light transmittance and color stability. Nonetheless, the high crystallinity and stability of YI8-P(3HB) are expected to preserve its mechanical properties.

## 4. Conclusions

Molasses is a widely available bioresource, with sucrose from sugarcane serving as a relatively inexpensive carbon source. However, the number of microorganisms capable of using sucrose as the sole carbon source is limited. In this study, we isolated a novel P(3HB)-producing strain, *Limimaricola* sp. YI8, which is capable of efficiently utilizing sucrose as a carbon source. Upon cultivating *Limimaricola* sp. YI8 under optimized conditions using sucrose as the sole carbon source, 6.2 g/L of P(3HB) was produced, accumulating P(3HB) granules at up to 58.8% of the DCW. Furthermore, using molasses as the carbon source, 5.2 g/L of P(3HB) was produced. Compared with other strains that produce P(3HB) using sucrose, this strain showed relatively high titers and P(3HB) content without any genetic engineering.

The P(3HB) film extracted from YI8 exhibited a distinct pinkish color, which was attributed to a naturally produced carotenoid dye. Determination of mechanical properties using UTM analysis showed that P(3HB) from *Limimaricola* sp. YI8 was relatively more rigid and less flexible than the P(3HB) films from *E. coli* and *C. necator* H16. In DSC analysis to determine the thermal properties, the P(3HB) of *Limimaricola* sp. YI8 exhibited a Tm value similar to that of the other P(3HB) films. Overall, this study presents a comprehensive exploration, encompassing the screening of new P(3HB)-producing strains to a detailed analysis of the properties inherent in the extracted P(3HB) films, and confirms that *Limimaricola* sp.YI8 is a promising P(3HB)-producing strain.

## Figures and Tables

**Figure 1 polymers-17-01471-f001:**
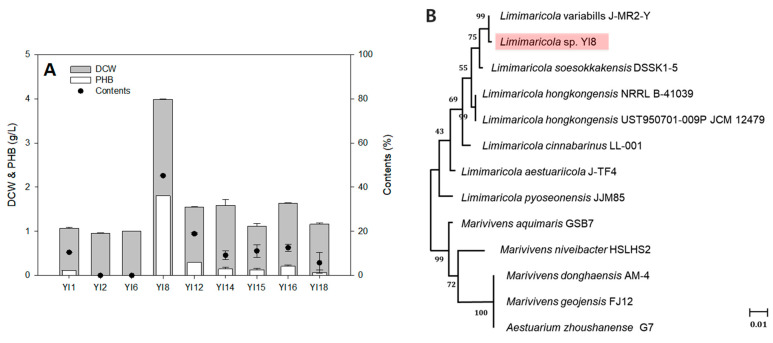
Screening the novel polyhydroxybutyrate P(3HB)-producing strain *Limimaricola* sp. YI8. (**A**) Screening P(3HB)-producing microbes from the marine environment. *Limimaricola* sp. YI8 produced 1.8 g/L of P(3HB) with 4 g/L dry cell weight (DCW). (**B**) Phylogenetic tree of *Limimaricola* sp. YI8.

**Figure 2 polymers-17-01471-f002:**
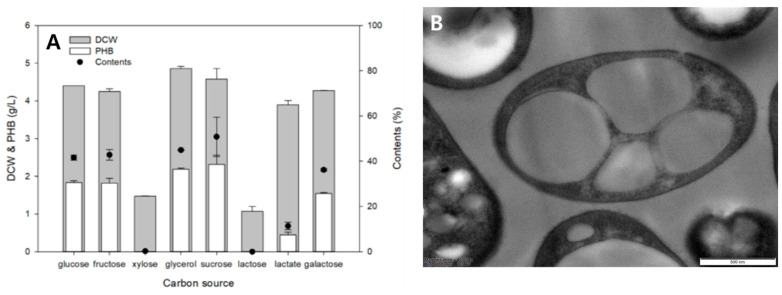
Carbon source test. (**A**) Optimization of carbon source. Cultivation was performed in 14 mL test tubes containing 5 mL of marine broth (MB) medium supplemented with 1% (*w*/*v*) of each sugar. After 1% (*v*/*v*) inoculation, the cultures were incubated at 30 °C and 200 rpm for 3 days. The highest P(3HB) production (2.31 g/L) was observed with sucrose as a carbon source. (**B**) Transmission electron microscopy image of *Limimaricola* sp. YI8.

**Figure 3 polymers-17-01471-f003:**
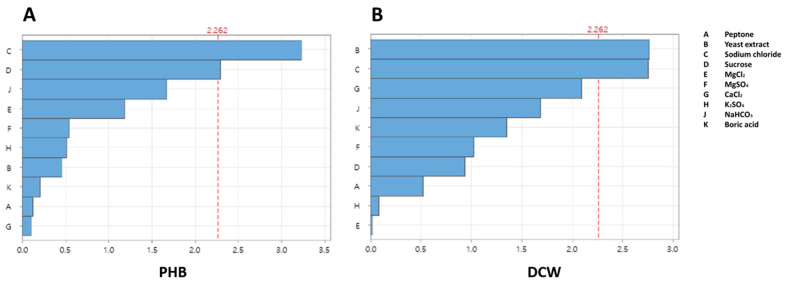
Pareto chart of the 10-factor standard effects on poly(3-hydroxybutyrate) and dry cell weight (DCW) production. (**A**) Pareto chart for P(3HB) production. (**B**) Pareto chart for DCW.

**Figure 4 polymers-17-01471-f004:**
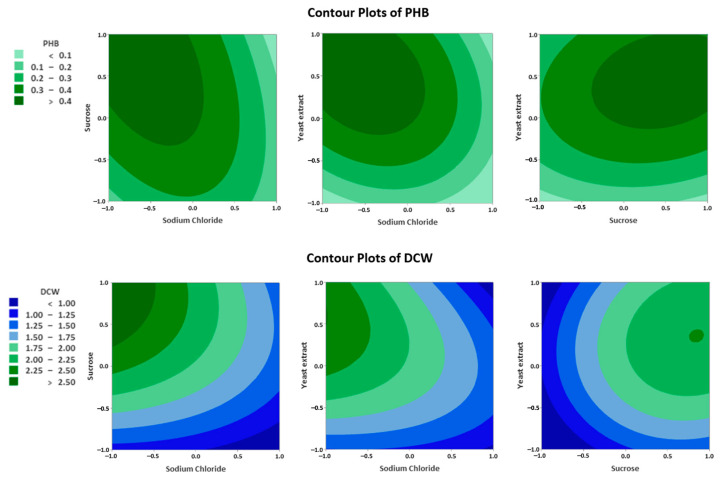
Contour plots showing the effects of independent variables on dry cell weight (DCW) and P(3HB) production.

**Figure 5 polymers-17-01471-f005:**
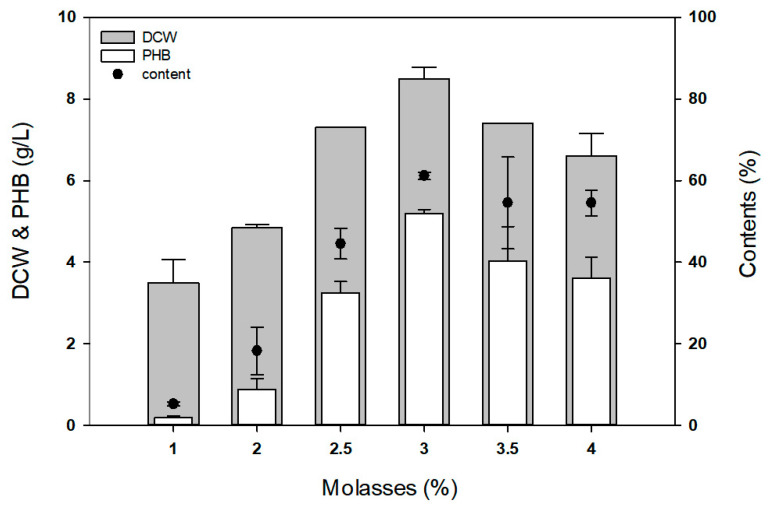
P(3HB) production using different concentration of molasses as the sole carbon source. DCW: dry cell weight.

**Figure 6 polymers-17-01471-f006:**
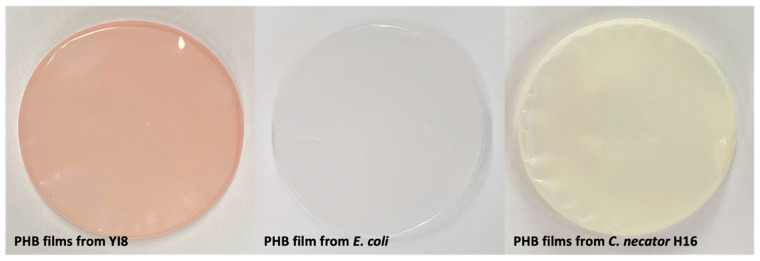
Formation of P(3HB)) films by extracting P(3HB) through solvent extraction. *E. coli*: *Escherichia coli*; *C. necator*: *Cupriavidus necator*.

**Table 1 polymers-17-01471-t001:** Previous studies on P(3HB) production from sucrose.

Strain	Culture	DCW (g/L)	P(3HB) (g/L)	P(3HB) Content (%)	Ref.
*Reunions naejangsanensis* BIO-TAS2-2	Batch	3.47	0.93	26.80	[15]
*Burkholderia sacchari* LFM 101	Batch	2.6540.076	0.879	33.125.989	[16]
*Burkholderia sacchari* DSM 17165	Fed Batch	70.0	36.8	53.0	[17]
*Halomonas boliviensis*	Batch	14	7.7	54.0	[18]
*Ralstonia eutropha* 437-540 (pKM212-SacCReAB)	Batch	0.590.01	0.340.01	56.00.7	[19]
*Bacillus megaterium* uyuni S29	Batch	12.350.34	−	50	[20]
*Bacillus megaterium* BA-019	Batch	7.05	3.8	55.46	[21]
*Bacillus endophyticus*	Batch	1.72	0.8	49.47	[22]
*Limimaricola* sp. YI8	Batch	11.1	6.2	55.86	This study

**Table 2 polymers-17-01471-t002:** Bacterial strains and plasmids used in this study.

Strain or Plasmid	Description	Ref.
Bacterial strain		
*Limimaricola* sp. YI8		This study
*Escherichia coli* KSYH(DE3)	ΔaraBAD, ΔrhaBAD, BW25113 (DE3) derivative	[25]
*Cupriavidus necator* H16	Gram-negative, Widely used as a model organism for PHA biosynthesis studies	Laboratory stock
Plasmids		
pLW487	pEP2-based plasmid (SpecR) carrying *bktB*, *phaB*, and *phaC* from *R. eutropha* under the trc promoter.	[26]

**Table 3 polymers-17-01471-t003:** GPC and UTM analysis of P(3HB) films produced by *Limimaricola* sp. YI8, *Escherichia coli* KSYH, and *Cupriavidus necator* H16.

		*Limimaricola* sp. YI8	*E. coli* KSYH::pLW487	*C. necator* H16
GPC	Mn (×10^6^)	1.09 ± 0.06	1.37 ± 0.02	1.27 ± 0.03
	Mw (×10^6^)	1.29 ± 0.05	1.58 ± 0.07	1.6 ± 0.03
	PDI (Mw/Mn)	1.24 ± 0.03	1.15 ± 0.04	1.28 ± 0.01
UTM	Tensile strength (MPa)	32.5	36.4	22.1
	Elongation at break (%)	3	7.2	12.9
	Young’s modulus (MPa)	1931.9	1337.6	701.7

**Table 4 polymers-17-01471-t004:** DSC analysis of P(3HB) films produced from *Limimaricola* sp. YI8, *Escherichia coli*, and *Cupriavidus necator*.

Strain	Tc (°C)	Tm (°C)
*Limimaricola* sp. YI8	106.2	172.2
*Escherichia coli* KSYH::pLW487	112.4	175.9
*Cupriavidus necator* H16	84.7	172.9

## Data Availability

Data will be made available on request.

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
