# Peer review of "Sucrose-Based Screening of a Novel Strain, Limimaricola sp. YI8, and Its Application to Polyhydroxybutyrate Production from Molasses"

_polymers, 2025, doi:10.3390/polym17111471_

Round 1

Reviewer 1 Report

Comments and Suggestions for Authors

Polymers (ISSN 2073-4360)

Manuscript ID :polymers-3615866

Manuscript Title: Sucrose-based screening of a novel strain, Limimaricola sp. YI8, and its application to polyhydroxybutyrate production from molasses

Dear Editor,

In this research, Limimaricola sp. YI8 strain was used for the production of Polyhydroxybutyrate (PHB) with high biodegradation capacity.Plastic pollution is one of the most important environmental pollutants of recent years, and the polyethylene compounds that make up plastics accumulate in natural resources and living tissues, causing many diseases, including cancer. Using degradable plastic instead of plastics that are difficult to break down can be an environmentally friendly practice.

However, it is not economically possible to produce plastic for mass use using microbial methods. In this respect, the scientific aspect of the study can be taken into consideration rather than its economic aspect.

Although this article is not economical, it can be accepted for scientific publication.

Although this article is not economical, it can be accepted for publication from a scientific point of view. It can be accepted as a medium-level publication.

The purpose of the research, the production method of polyhydroxybutyrate (PHB) and its areas of use should be explained in more detail. Therefore, I recommend minor revisions.

Best wishes and regards

Author Response

Comments 1:

Dear Editor,

In this research, Limimaricola sp. YI8 strain was used for the production of Polyhydroxybutyrate (PHB) with high biodegradation capacity. Plastic pollution is one of the most important environmental pollutants of recent years, and the polyethylene compounds that make up plastics accumulate in natural resources and living tissues, causing many diseases, including cancer. Using degradable plastic instead of plastics that are difficult to break down can be an environmentally friendly practice.

However, it is not economically possible to produce plastic for mass use using microbial methods. In this respect, the scientific aspect of the study can be taken into consideration rather than its economic aspect.

Although this article is not economical, it can be accepted for scientific publication.

Although this article is not economical, it can be accepted for publication from a scientific point of view. It can be accepted as a medium-level publication.

The purpose of the research, the production method of polyhydroxybutyrate (PHB) and its areas of use should be explained in more detail. Therefore, I recommend minor revisions.

Best wishes and regards

Response 1: 

We sincerely appreciate the reviewer’s thoughtful comments. While we agree that industrial-scale production of PHB still faces economic challenges, we would like to emphasize that the primary goal of this study was not to demonstrate economic feasibility, but to identify and characterize a novel wild-type strain capable of utilizing low-cost carbon sources such as molasses for PHB production.

Unlike genetically engineered strains, Limimaricola sp. YI8 shows significant PHB accumulation from molasses without any prior metabolic engineering. We believe that this finding provides valuable insights for future research on strain improvement and resource valorization. In the revised manuscript, we have clarified this point and highlighted the scientific relevance of YI8 as a promising natural producer.

Reviewer 2 Report

Comments and Suggestions for Authors

The paper «Sucrose-based screening of a novel strain, Limimaricola sp. YI8, and its application to polyhydroxybutyrate production from molasses» is devoted to the study of PHA production by new strain Limimaricola sp. YI8 using molasses as sole carbon source. The article corresponds to the scope of the Polymers. However, it needs to address some comments, and thus require substantial major revision to improve the quality of the manuscript. It is necessary to correct the paper, the research methods should be expanded and clarified (especially the cultivation process should be written more clearly and distinctly).

  1. Replace PHB with P(3HB) throughout the text.
  2. Lines 28-29. It would be more correct to write that you extracted the polymer (but not the PHB film) and obtained a film from it. Check throughout the text.
  3. Lines 29-30. A universal test machine is used to determine the mechanical properties. Differential scanning calorimetry is used to determine the temperature characteristics of polymers.
  4. 2.3. Section. Describe the cultivation conditions in more detail (what flasks were used, what kind of shaker-incubator was used, the composition of the medium, etc.).
  5. 2.3. Section. Where are the data on the cultivation of the studied strain at different temperatures (25, 30 and 37 °C)?
  6. 2.8. Section. Add information about the molasses used in the work (manufacturer, composition).
  7. Figure 2. Add information about electron microscopy (conditions, device) to the Materials and Methods Section.
  8. Figure 2A. Add detailed information about the cultivation conditions
  9. 3.4 Section. Add detailed information about the cultivation conditions. Did you measure the residual sugar content in the medium at the end of the cultivation? The higher biomass content on molasses may be due to a higher concentration of sugars than in the experiment with sucrose (Figure 2A). Were the same concentrations of sugars used in these experiments?
  10. 3.4 Section. Add information about the E. coli and C. necator strains to the Materials and Methods section (where these strains were obtained, cultivation conditions, composition of the medium, etc.).
  11. Figure 6. Apparently, the colored films could be due to insufficient purification of the polymer from impurities (lipids, pigments, etc.). To obtain a polymer free of impurities, the polymer is usually re-dissolved with subsequent precipitation to obtain a polymer free of impurities (even repeatedly). Since, it is known that impurities can affect the temperature characteristics and mechanical properties of polymers.
  12. Usually, melting point is used, not melting temperature.
  13. Lines 375-376. The sentence should be deleted, since the article does not contain data on photodegradation resistance under UVA.

Reviewer 3 Report

Comments and Suggestions for Authors

Lee et. al have described here the identification of a new microbe species that utilizes sucrose as a carbon source to produce polyhydroxybutyrate (PHB). Up to 6.2 g/L PHB were produced from sucrose sole carbon source, and their thermal and mechanical properties were shown to be comparable with PHBs produced from other microbes. This article is well written and represents a solid contribution to the field. Therefore, this reviewer would recommend acceptance of the article.  The only minor comment is that the reference format is not always consistent and some of them are missing article number/page number/doi, i.e. reference 1 and 8. 

Author Response

Comments 1: Lee et. al have described here the identification of a new microbe species that utilizes sucrose as a carbon source to produce polyhydroxybutyrate (PHB). Up to 6.2 g/L PHB were produced from sucrose sole carbon source, and their thermal and mechanical properties were shown to be comparable with PHBs produced from other microbes. This article is well written and represents a solid contribution to the field. Therefore, this reviewer would recommend acceptance of the article.

The only minor comment is that the reference format is not always consistent and some of them are missing article number/page number/doi, i.e. reference 1 and 8.

Response 1: Thank you for your help with our paper. We corrected the reference format as you mentioned.

Round 2

Reviewer 2 Report

Comments and Suggestions for Authors

Paper can be accepted in present form